# Maternal and congenital toxoplasmosis in Mayotte: Prevalence, incidence and management

**Laure Kamus**[1,2]\*, **Sophie Belec**[3], **Laurent Lambrecht**[4], **Soumeth Abasse**[4], **Sophie Olivier**[5], **Patrice Combe**[5], **Pierre-Emmanuel Bonnave**[6], **Christelle Vauloup-Fellous**[7]

**1** Department of Medical Biology, Félix-Guyon Hospital Center, Saint-Denis, La Réunion, France, **2** UMR Processus Infectieux en Milieu Insulaire Tropical (PIMIT), CNRS 9192, INSERM U1187, IRD 249, Université de La Réunion, Saint-Denis, La Réunion, France, **3** Department of Obstetrics and Gynaecology, Mayotte Hospital Center, Mayotte, France, **4** Paediatric Centre, Mayotte Hospital Centre, Mayotte, France, **5** Department of Medical Biology, Mayotte Hospital Centre, Mayotte, France, **6** Mayo Bio Laboratory, Mamoudzou, Mayotte, France, **7** Universite Paris Saclay, INSERM U1193, AP-HP, Hôpital Paul Brousse, Virology Department, Villejuif, France

\* laure.kamus@chu-reunion.fr

**Data Availability Statement:** All relevant data are within the manuscript and its Supporting Information files.

## Abstract

### Background

Toxoplasmosis is an infection caused by an intracellular protozoan, *Toxoplasma gondii*. It is usually asymptomatic, but toxoplasmosis acquired during pregnancy can cause congenital toxoplasmosis, potentially resulting in fetal damage. Epidemiological information is lacking for toxoplasmosis in Mayotte (a French overseas territory). We evaluated (1) the prevalence of maternal toxoplasmosis, (2) the incidence of maternal and congenital toxoplasmosis, and (3) the management of congenital toxoplasmosis in Mayotte.

### Methodology / Principal Findings

We collected all the available data for toxoplasmosis serological screening during pregnancy and maternal and congenital cases of toxoplasmosis obtained between January 2017 and August 2019 at the central public laboratory of Mayotte (Mamoudzou). Using toxoplasmosis serological data from samples collected from 16,952 pregnant women we estimated the prevalence of toxoplasmosis in Mayotte at 67.19%. Minimum maternal toxoplasmosis incidence was estimated at 0.29% (49/16,952, 95% CI (0.0022–0.0038)), based on confirmed cases of maternal primary infection only. The estimated incidence of congenital toxoplasmosis was 0.09% (16/16,952, 95% CI (0.0005–0.0015). Missing data made it difficult to evaluate management, but follow-up was better for mothers with confirmed primary infection and their infants.

### Conclusions / Significance

The seroprevalence of toxoplasmosis among pregnant women and the incidence of toxoplasmosis are higher in Mayotte than in mainland France. There is a need to improve the antenatal toxoplasmosis screening and prevention programme, providing better

**Funding:** The authors received no specific funding for this work.

**Competing interests:** The authors have declared that no competing interests exist.

information to physicians and the population, to improve management and epidemiological monitoring.

## Author summary

Toxoplasmosis, a zoonotic disease caused by a ubiquitous protozoan parasite *Toxoplasma gondii* is usually asymptomatic. However, maternal infection during pregnancy can cause congenital toxoplasmosis with potentially severe neurologic or ophthalmological sequelae. Prevalence varies from country to country depending on socio-economic conditions, dietary habits and hygiene and the prevalence is higher in resource-poor countries. In Mayotte (a French overseas territory), recommendations for toxoplasmosis prevention and screening are partially applied. Indeed, many women don't speak French, serological follow up is generally irregular and responsible for delayed introduction of treatment and consequently more congenital toxoplasmosis cases. To our knowledge, few published data were available on maternal and congenital toxoplasmosis for this territory. A better understanding of the epidemiology of this parasite would allow surveillance, information and limitation of maternal-fetal transmission to reduce the number of congenital toxoplasmosis cases and improve the management of toxoplasmosis in Mayotte.

## Introduction

*Toxoplasma gondii* is a ubiquitous protozoan parasite that causes *Toxoplasmosa* infection, a zoonosis with a worldwide distribution. This intracellular parasite infects warm-blooded animals, including humans, and is estimated to have infected more than one third of the world's population [1]. Primary infection is usually (80% cases) asymptomatic in immunocompetent individuals [2,3]. However, if a woman is infected during pregnancy, toxoploasmosis may occur, with potentially severe sequelae (neurological and/or ocular damage) or a fatal outcome (*in utero* abortion, foetal/neonatal death) [1,4,5]. The risk of transmission to the foetus is closely related to gestational age at infection in the mother, ranging from <15% in the first trimester of pregnancy to almost 70% in the third trimester [6]. By contrast, congenital Toxoplasmosis (CT) is most severe if the mother is infected during the first trimester of pregnancy [3,5]. The rapid, early diagnosis of infection in the mother makes it possible to provide treatment which has been shown to lower transplacental transmission rates, and specific follow-up: regular ultrasound examinations, amniocentesis and, if necessary, a change of treatment [7,8]. In France, hygiene recommendations have been implemented since 1983 and, since 1992 it is mandatory to perform monthly serological screening in seronegative women and to provide hygiene recommendations until one month after delivery [9,10]. In 2006, a surveillance programme for CT was launched in France (mainland France and overseas territories), providing valuable information annually concerning the prevalence and management of these infections [11]. Mayotte is an island that became an overseas *département* (a territorial unit similar to a county) of France in 2011; it was not included in the surveillance network until 2020 [12]. However, French recommendations for screening and prevention strategies in all pregnant women were applied in Mayotte for several years before it acquired this status.

In 2019, Mayotte had 270,000 inhabitants, with about 9,600 births recorded by the regional health agency each year. Seroprevalence in mainland France has been decreasing since the 1960s, and is currently estimated at about 30% [11,13]. Little information about seroprevalence is available for Mayotte, and the real figures on the ground may be very different from those

for mainland France, because of the tropical weather conditions, which may facilitate the survival of oocysts in the environment (hot and humid climate, long periods of rainfall) and high rates of reservoir infection (favoured by the large number of young feral cats). Moreover, this island has a high population density, with many living in precarious conditions, high rates of poverty and illiteracy, and a diet very different from that in mainland France [14]. The prevalence of toxoplasmosis could therefore be higher in Mayotte than in mainland France, and studies are required to determine the actual prevalence in Mayotte. Medical cover is insufficient in Mayotte (the density of doctors in Mayotte is one third that in mainland France for general practitioners, and one quarter that in mainland France for specialists) [15], and there are only two medical laboratories on the island (one private and one public laboratory). This lack of medical facilities undoubtedly has an impact on the management of pregnant women and their babies. We performed a retrospective study to evaluate the prevalence of maternal toxoplasmosis in Mayotte, the incidence of maternal and congenital *Toxoplasma* infection, and their management.

## Materials and methods

### Population

All pregnant women for whom at least one anti serological test for IgG and IgM was performed between January 2017 and August 2019 were included regardless of the gestational age at sampling.

Women with no record of pregnancy in the medical record were excluded as well as women with postpartum seroconversion.

### Data collection

Data were collected retrospectively from the patients' medical records. The following items were recorded for the women: all *Toxoplasma* serological/PCR results available, gestational age at maternal infection, gravidity, parity, amniocentesis/no amniocentesis, ultrasound examinations, and treatment (molecules, duration). The following items were recorded for the neonate: all *Toxoplasma* serological/PCR results available, cerebral ultrasound and ophthalmological examinations, and treatment at birth (molecules, duration).

Biological data were collected from the database of the central laboratory of the *Centre Hospitalier de Mayotte* (InLog, *INLOG* HOLDINGS FRANCE SAS, Limonest, France). In cases of missing data during pregnancy follow-up, we obtained the missing data from the only other laboratory in Mayotte (SELARL MAYO BIO, Mamoudzou), whenever possible.

### Serology and PCR

IgG and IgM were performed with Access TOXO (Beckman Coulter, Tennessee, USA), and IgG avidity tests were performed with VIDAS TOXO AVIDITY (BioMérieux, Craponne, France), according to the manufacturers' recommendations. Equivocal IgG results were considered as negative. High IgG avidity made it possible to exclude recent maternal infection within the last four months. In cases of low or intermediate IgG avidity, it was not possible to exclude recent infection, and IgG kinetics were analysed on samples collected three to four weeks later.

PCR (on amniotic fluid and blood) was performed at a private laboratory in the Parisian region (CERBA, St Ouen l'Aumône, France).

### Case definition (according to French National recommendations (9, 21, 22))

**Confirmed maternal toxoplasmosis.** maternal toxoplasmosis was considered certain if seroconversion occurred during pregnancy, or if positive results were obtained for the

detection of IgM and IgG, with low IgG avidity and a doubling of IgG titre on a second sample collected three to four weeks later in women less than two months into gestation at the time of the first serological test.

**Possible maternal *Toxoplasma* infection.** maternal *Toxoplasma* infection was considered possible if positive results were obtained for IgM and IgG detection, with low or intermediate IgG avidity, and a stable IgG titre for a second sample collected three to four weeks later in women less than two months into gestation at time of the first serological test.

**Non-excluded maternal *Toxoplasma* infection.** maternal *Toxoplasma* infection could not be excluded if positive results were obtained for IgM and IgG detection, with high IgG avidity, in women more than four months into gestation.

**Confirmed CT.** CT was considered certain if positive results were obtained for IgM detection and/or if PCR results on peripheral blood were positive in the first week of life.

**Seroprevalence of *Toxoplasma* infection calculation.** Seroprevalence was calculated dividing the number of pregnant women with IgG above the cut off at their first test by the total number of pregnancies investigated.

**Incidence of *Toxoplasma* infection calculation.** Incidences were calculated as follows:

- Minimal incidence was calculated dividing the number of confirmed maternal toxoplasmosis cases by the total number of pregnancies investigated regardless of the length of pregnancy covered by the screening;

- Maximal incidence was calculated diving the number of confirmed added by the number of possible maternal *Toxoplasma* infection cases by the total number of pregnancies investigated regardless of the length of pregnancy covered by the screening.

- Minimal incidence of CT was calculated diving the number of confirmed CT cases by the total estimated number of infants borned to the pregnancies investigated.

## Statistical analysis

Univariate analysis was performed to compare the incidence of maternal infection and CT between Mayotte and mainland France and/or other French overseas territories. A z-test was performed with Stata software (version 12.0), and *p*-values <0.05 were considered significant. A 95% confidence interval (95% CI) was calculated with the free software VassarStats: Website for statistical Computation on http://vassarstats.net/.

## Results

The regional health agency (ARS) of Mayotte recorded approximately 9,600 births each year from 2017 to 2019. During our study period (January 2017 to August 2019), an estimated 25,745 infants were born alive to 25,088 women [16,17]. We investigated 16,952 pregnancies, and our study therefore had an exhaustivity of 67.57% (95% CI: 0.67–0.68). The demographic records provided by the Mayotte ARS indicated that 17,240 infants were born to the 16,952 mothers for whom serological results were available.

## Prevalence of maternal toxoplasmosis

From January 2017 to August 2019, IgG tests were positive in 67.19% (11,390/16,952, 95% CI (0.66–0.68)) of pregnant women and negative in 31.77% (5,386/16,952, 95% CI (0.31–0.32)) of pregnant women. In 1.04% (176/16,952, 95% CI (0.009–0.012)) of cases IgG results were equivocal, and the patients were considered seronegative in subsequent analyses. The prevalence of maternal toxoplasmosis in Mayotte was, therefore, 67.19%.

### Incidence of maternal toxoplasmosis

During the study period, IgM was detected in 272 patients, two of whom were excluded due to seroconversion during the postpartum period rather than during pregnancy.

The results for the 270 samples testing positive for IgM collected from pregnant women were (Fig 1):

- 46/270 (17.78% (95% CI (0.14–0.23)) tested negative for IgG, and 27/46 (58.70%) subsequently seroconverted and were considered to be cases of confirmed toxoplasmosis;

- 224/270 (82.96%, 95% CI (0.78–0.87)) tested positive for IgG. In 59/224 (26.34%, 95% CI (0.21–0.32)) cases, IgG avidity was low. In 22/224 (9.82%, 95% CI (0.07–0.14)) cases, an analysis of IgG kinetics confirmed confirmed toxoplasmosis. In 6/224 cases, preconception infection was considered. Postconceptional infection was considered possible in 31/224 cases (0.14%, 95% CI (0.10–0.19)) because the first sample was collected more than two months into the pregnancy;

- 165/224 (73.66%, 95% CI (0.68–0.79)) tested positive for IgG with high IgG avidity. In 86/165 cases, preconceptional infection was considered. Postconceptional infection could not be excluded in 79/165 (47.88%, 95% CI (0.40–0.55)) cases, because the first sample was collected more than four months into the pregnancy.

The minimum incidence of maternal infection (considering only confirmed toxoplasmosis) was 0.29% (49/16,952, 95% CI (0.0022–0.0038)) and the maximal incidence (considering

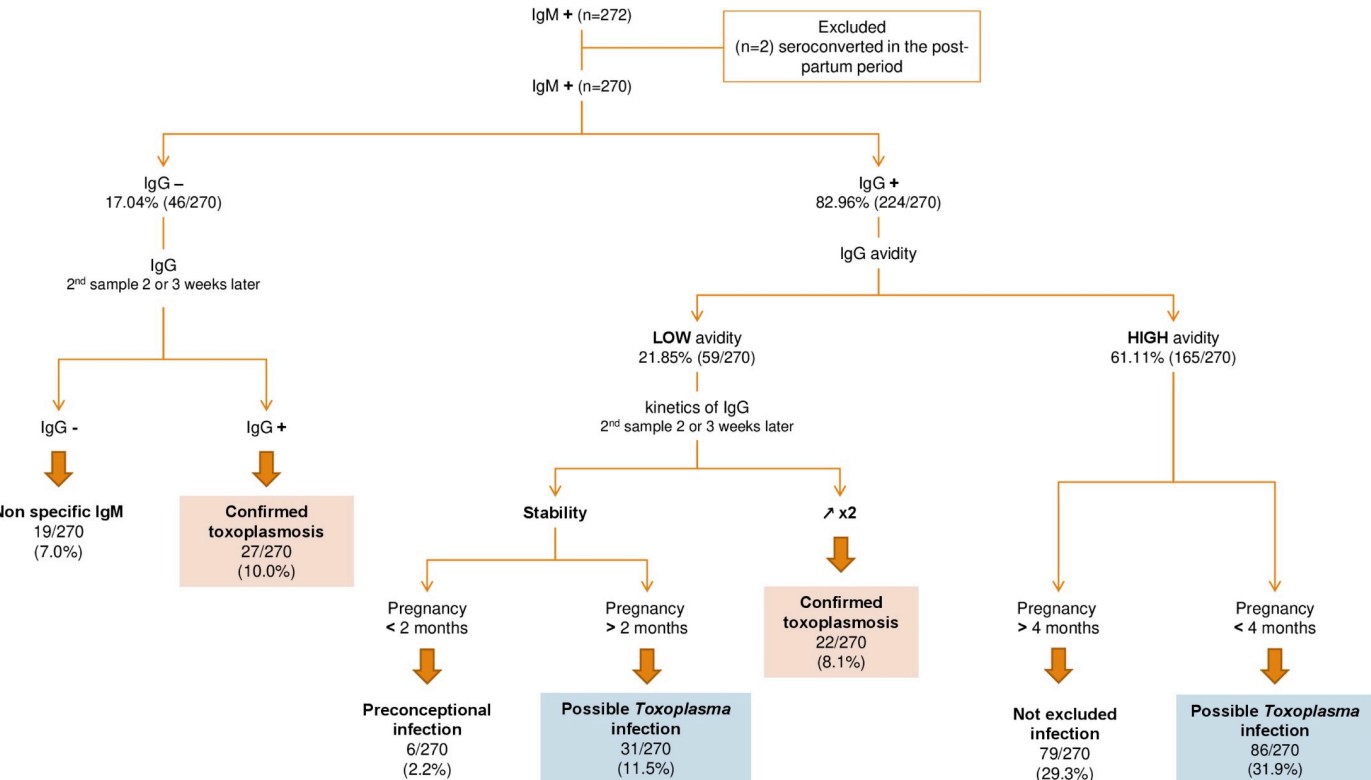

**Fig 1. Classification of maternal toxoplasmosis with primary infection.** Classification according to the recommendations of the French National Reference Centre for Toxoplasmosis [9,18,19].

confirmed toxoplasmosis and possible *Toxoplasma* infections) was 0.47% (80/16,952, 95% CI (0.0038–0.0058)). These values correspond to 2.89 to 4.72/1,000 pregnant women per year. *Toxoplasma* infection during pregnancy could not be excluded in 79/270 cases (0.29%, 95% CI (0.24–0.35).

### Demographic information and management of maternal infection

Focusing on the 49 confirmed toxoplasmosis, most of these cases occurred during the third trimester of pregnancy (65.31%, 32/49, 95% CI (0.51–0.77)), treatment information was not available for 53.06% (26/49, 95% CI (0.39–0.66) and only 32.65% (16/49, 95% CI (0.21–0.47)) received treatment (spiramycin, as Rovamycine) (Table 1). Prenatal ultrasound follow-up was performed in 48.98% (24/49) cases, and abnormalities related to CT were observed in 37.50% (9/24). Amniocentesis was performed in only 4.08% (2/49) of the women (Table 1). The demographic and management details for possible *Toxoplasma* infections and non-excluded cases are available in Table 1.

### Incidence of congenital toxoplasmosis

In total, 50 infants were born to the 49 mothers with confirmed toxoplasmosis. During the study period, CT diagnosis was based on positive IgM and/or positive PCR results for a peripheral blood sample collected from the neonate. At least one serum sample was collected from most neonate (90.00% (45/50)) during the neonatal period (median: 1 day (range 1–54 days)) and PCR was also performed on blood for 78.00% (39/50) (Table 2). Overall, 16/50 (32.00%) children had CT (8 males and 8 females) (Fig 2).

**Table 1. Demographic and management data for pregnant women with confirmed toxoplasmosis, possible *Toxoplasma* infection and non-excluded primary infection.** * Abnormalities related to congenital toxoplasmosis.

| | | | Confirmed toxoplasmosis (n = 49)(%) | Possible Toxoplasma infections (n = 31)(%) | Non-excluded (n = 79)(%) |
|---|---|---|---|---|---|
| *Age* | | | 25 (16–38) | 27 (15–54) | 26 (15–41) |
| | Unknown | | 4/49 (8.16) | 3/31 (9.68) | 2/79 (2.53) |
| *Gravidity* | | | 3 (1–8) | 2 (1–6) | 6 (1–8) |
| | Unknown | | 22/49 (44.90) | 18/31 (58.06) | 29/79 (37.71) |
| *Parity* | | | 5 (1–6) | 3 (1–4) | 8 (1–7) |
| | Unknown | | 22/49 (44.90) | 19/31 (61.29) | 30/79 (37.97) |
| *Trimester of infection* | 1st trimester | | 0/49 (0.00) | 0/31 (0.00) | 1/79 (1.27) |
| | 2nd trimester | | 15/49 (30.61) | 12/31 (38.71) | 4/79 (5.06) |
| | 3rd trimester | | 32/49 (65.31) | 16/31 (51.61) | 71/79 (89.87) |
| | Unknown | | 2/49 (4.08) | 3/31 (9.68) | 3/79 (3.80) |
| *Treatment* | Yes | | 16/49 (32.65) | 3/31 (9.680) | 1/79 (1.27) |
| | No | | 7/49 (14.29) | 0/31 (0.00) | 0/79 (0.00) |
| | Unknown | | 26/49 (53.06) | 28/31 (90.32) | 78/79 (98.73) |
| *Antenatal ultrasound follow-up* | Yes | | 24/49 (48.98) | 2/31 (6.45) | 1/79 (1.27) |
| | | Normal | 15/24 (62.50) | 1/2 (50.00) | 1/1 (100.00) |
| | | Abnormal* | 9/24 (37.50) | 1/2 (50.00) | 0/1 (0.00) |
| | No | | 24/49 (48.98) | 0/31 (0.00) | 4/79 (5.06) |
| | Unknown | | 1/49 (2.04) | 29/31 (93.55) | 74/79 (93.67) |
| *Amniocentesis* | Yes | | 2/49 (4.08) | 1/31 (3.23) | 1/79 (1.27) |
| | No | | 47/49 (95.92) | 30/31 (96.77) | 78/79 (98.73) |

**Table 2. Biological data, treatment and imaging information for infants born to women with confirmed toxoplasmosis, possible *Toxoplasma* infection or non-excluded primary infection.** Two sets of twins were born, one to a mother with confirmed toxoplasmosis and the other to a mother with non-excluded primary infection.

| | | *Infants born to mothers with confirmed toxoplasmosis (n = 50)(%)* | *Infants born to mothers with possible Toxoplasma infection (n = 31)(%)* | *Infants born to mothers with non-excluded primary infection (n = 80)(%)* |
|---|---|---|---|---|
| *Serological tests* | Yes | 45/50 (90.00) | 7/31 (22.58) | 2/80 (2.50) |
| | No | 5/50 (10.00) | 0/31 (0.00) | 1/80 (1.25) |
| | Unknown | 0/50 (0.00) | 24/31 (77.42) | 77/80 (96.25) |
| *PCR* | Yes | 39/50 (78.00) | 9/31 (29.03) | 3/80 (3.75) |
| | No | 11/50 (22.00) | 0/31 (0.00) | 4/80 (5.00) |
| | Unknown | 0/50 (0.00) | 22/31 (70.97) | 73/80 (91.25) |
| *Treatment* | Yes | 10/50 (20.00) | 1/31 (3.23) | 0/80 (0.00) |
| | No | 2/50 (4.00) | 0/31 (0.00) | 1/80 (1.25) |
| | Unknown | 38/50 (76.00) | 30/31 (96.77) | 79/80 (98.75) |
| *Imaging (antenatal ultrasound)* | Yes | 25/50 (50.00) | 2/31 (6.45) | 1/80 (1.25) |
| | No | 24/50 (48.00) | 0/31 (0.00) | 4/80 (5.00) |
| | Unknown | 1/50 (2.00) | 29/31 (93.55) | 75/80 (93.75) |

In total, 31 infants were born to the 31 mothers with possible *Toxoplasma* infections, and 80 were born to the 79 mothers with non-excluded primary infection. Only 9/111 children underwent at least one serological test and 12/111 underwent PCR on blood (Table 2).

Given the large numbers of missing data for children born to women with possible *Toxoplasma* infections and non-excluded primary infections, only a minimum incidence of CT was

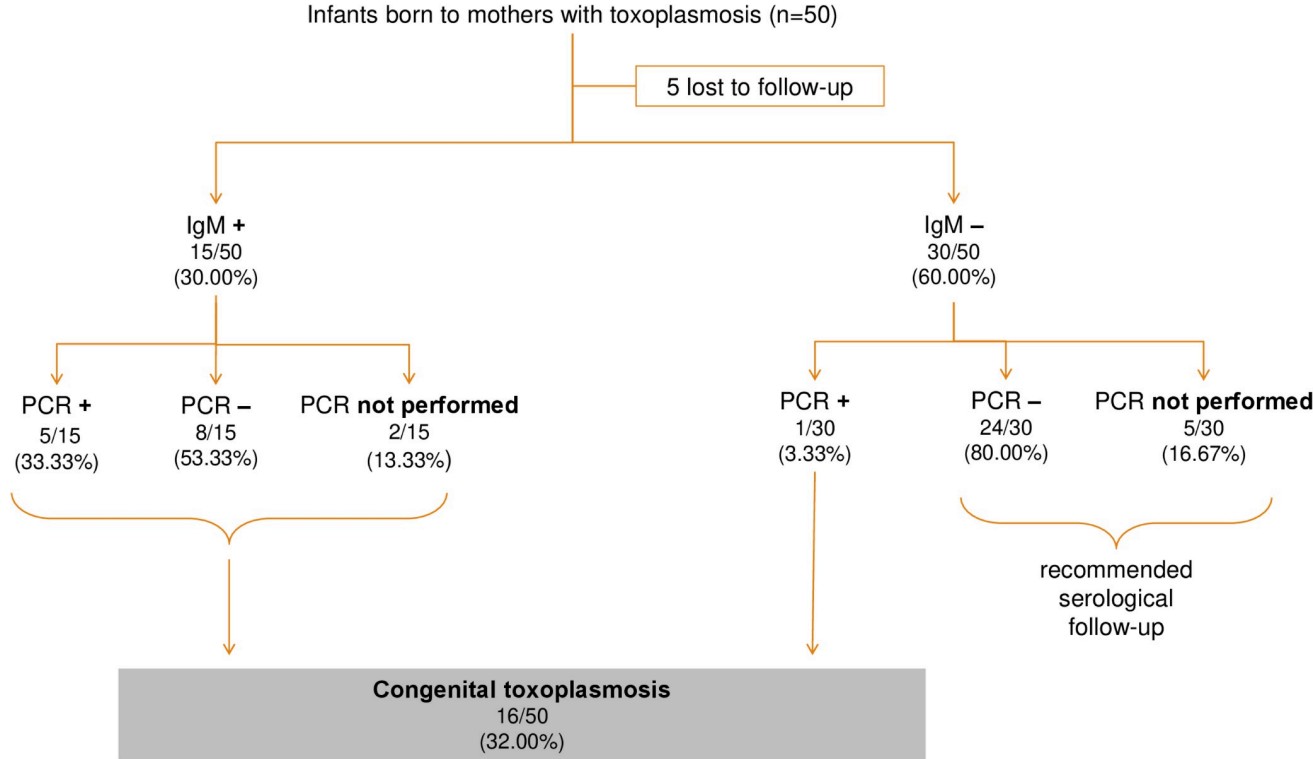

**Fig 2. Diagnosis algorithm for congenital toxoplasmosis.** PCR: polymerase chain reaction.

**Table 3. Management of newborns with confirmed congenital infection (CT).** *Abnormalities related to congenital toxoplasmosis. (Imaging abnormality: one case with small subependymal haemorrhages typical of toxoplasmosis (no calcification and no ventricular dilation) confirmed on check-up imaging and one case of persistent microcysts in the caudate nucleus).

| | | | Congenital infections (IgM + and/or PCR +) (n = 16)(%) | Newborn's gender (males/females) (n = 16) |
|---|---|---|---|---|
| **Serological tests and PCR** | Yes | | 16/16 (100.00) | 8/8 |
| | No | | 0/16 (0.00) | 0/0 |
| | Unknown | | 0/16 (0.00) | 0/0 |
| **Antenatal ultrasound** | Yes | | 10/16 (62.50) | 5/5 |
| | | Normal | 6/10 (60.00) | 3/3 |
| | | Abnormal* | 4/10 (40.00) | 2/2 |
| | No | | 6/16 (37.50) | 3/3 |
| | Unknown | | 0/16 (0.00) | 0/0 |
| **Trans-fontanelle cerebral ultrasound scan** | Yes | | 10/16 (62.50) | 5/5 |
| | | Normal | 8/10 (80.00) | 4/4 |
| | | Abnormal* | 2/10 (20.00) | 1/1 |
| | No | | 2/16 (12.50) | 1/1 |
| | Unknown | | 4/16 (25.00) | 2/2 |
| **Eye fundus examination** | Yes | | 7/16 (43.75) | 2/5 |
| | | Normal | 7/7 (100.00) | 2/5 |
| | | Abnormal* | 0/7 (0.00) | 0/0 |
| | No | | 0/16 (0.00) | 0/0 |
| | Unknown | | 9/16 (56.25) | 6/3 |
| **Follow-up > 1 month** | Yes | | 5/16 (31.25) | 1/4 |
| | No | | 0/16 (0.00) | 0/0 |
| | Unknown | | 11/16 (68.75) | 7/4 |

calculated, based on the demographic records of children born to women with confirmed toxoplasmosis. We estimated that 17,240 infants were born to the 16,952 mothers with available serological results during the study period. The minimal incidence was, therefore, 9.28/10,000 liveborn infants (16/17,240, 0.09% (95% CI (0.0005–0.0015)).

## Management of newborns

For infants born to mothers with confirmed toxoplasmosis, treatment information was available for 24.00% (12/50) cases and imaging data were available for 98.00% (49/100). More than 90.00% of the data were not reported in medical records (serological tests, PCR, treatment and imaging data) for infants born to mothers with possible *Toxoplasma* infections or non-excluded primary infections (Table 2).

Less than half the infants with CT (43.75%, 7/16) underwent eye fundus examination, and the results were normal in all cases (Table 3). More than half underwent transfontanelle cerebral ultrasound examinations (62.50%, 10/16), the results of which were normal in 80.00% of cases (8/10) (Table 3). Cerebral abnormalities were reported in two cases: a persistence of microcysts in the caudate nucleus in one infant, and small subependymal haemorrhages typical of toxoplasmosis in the second infant, with no calcifications or cerebral ventricle dilation in either case.

## Discussion

Due to epidemiological changes reduced opportunities to encounter parasites and thanks to several national prevention programmes, the prevalence of maternal toxoplasmosis has steadily decreased over the last few decades in mainland France. Indeed, it fell from 84% in 1960 to 33.7% in 2016 [9,12,20–25]. A similar decrease has been reported for French overseas territories, despite less data being available. The first studies in French overseas territories in 1995 reported a seroprevalence of 61.4%, which decreased to 42.9% in 2016 (French West Indies, French Guiana, La Reunion) [20,24,25]. Prevalence increases with age and depends on geographic location, socioeconomic level, eating habits and cultural practices. In our study, in Mayotte, we found a maternal toxoplasmosis seroprevalence of 67.19%, a value similar to that reported for mainland France in the 1980s and higher than the values for other French overseas territories in 2016. One of the main factors underlying this difference is probably diet, particularly as concerns the consumption of meat and contaminated vegetables. Toxoplasmosis prevalence is lower in countries in which meat is eaten well cooked (United Kingdom, North America). It is slightly higher in mainland France, where meat is often eaten raw or smoked [9]. In Mayotte, meat is also often eaten raw or smoked, or cooked on a barbecue, which does not destroy toxoplasmosis cysts. Another important factor influencing prevalence is socioeconomic conditions, including, in particular, access to clean water, and proximity to animals (especially feral cats) that can serve as major reservoirs of *Toxoplasma gondii*. Hot and humid climatic conditions also favour the spread of toxoplasmosis [26,27]. One of the strengths of our study is its exhaustivity of about 67% for the 2017/2019 period. However, prevalence may well be underestimated in this study, as at least 20% of women in Mayotte are thought to illegal immigrants, with an even lower socioeconomic status, and many of these women may not have undergone serological testing at all during their pregnancy.

The incidence of toxoplasmosis during pregnancy varies with prevalence in general population. The incidence during pregnancy is well-documented in mainland France, due to the mandatory serological screening programme. This incidence has decreased over time, reaching a predicted 1.6/1,000 in 2020 [8,21]. In Mayotte, we found the incidence of maternal infection to be 2.89 to 4.72/1,000 per year. This higher incidence of maternal infection is consistent with the prevalence of 67.19%, and is significantly different from that of mainland France (*p*-value <0.05), although the calculated prevalence and incidence are derived from an estimate based on modelling However, this rate is probably underestimated due to the large number of cases that could not be formally classified. Indeed, one of the limitations of our study is the inconsistency of serological follow-up in pregnant women. In many cases, only one serological test was performed during pregnancy, often more than four months after conception. This lack of follow-up is due to local factors, such as a lack of awareness in women of child-bearing age, the late diagnosis of pregnancy and frequent misunderstanding of the French language (French is the mother tongue of only 10% of the mothers in Mayotte). This last factor is almost certainly also a major reason for the lack of awareness concerning behaviours at risk for *Toxoplasma* infection and for women not following primary toxoplasmosis prevention rules.

For the management of maternal infection, our study highlights a greater acceptance of non-invasive examinations, such as ultrasound scans, rather than amniocentesis (in women with confirmed toxoplasmosis particularly: 49.98% (24/49; 95% CI (0.36–0.63) and 4.08% (2/49; 95% CI (0.01–0.14), respectively). PCR on amniotic fluid is an excellent tool for confirming foetal infection and ensuring optimal management of the child at birth, particularly in terms of reducing unnecessary treatment with possible adverse effects [28,29]. Several studies have reported a risk of pregnancy loss of <1% during amniocentesis procedures [30]. The poor acceptance of amniocentesis in Mayotte is a serious obstacle to antenatal and neonatal

diagnosis, representing a loss of opportunity in the management of this disease. It is probably related to cultural and religious factors. Providing women with better information about the disease, the various diagnostic tests and their benefits would raise awareness, potentially increasing the acceptability of amniocentesis.

The worldwide incidence of CT ranges from 1 to 30/10,000 live-born infants [9,31]. In mainland France, the incidence is about 2 to 3 cases/10,000, and this rate has remained stable since 2007 [11,12]. In Mayotte, this incidence was estimated at 9.28/10,000 liveborn infants, which is three times the rate in mainland France (*p*-value <0.05). This difference may, of course, be due to the higher incidence of maternal infection, but it may also reflect a late introduction of treatment due to irregular and/or delayed serological follow-up, which concerns more than just toxoplasmosis. One limitation of our results was the significant number of missing data, but this lack of information may suggest that treatments were not prescribed or not taken by pregnant women. Most infants born to mothers with possible *Toxoplasma* infections and non-excluded primary infections were not explored, suggesting that the incidence reported here may be an underestimate. In new-borns, IgM tests were performed at the age of three days (mean: 2.87 days; median: 1 day; range: 0–43 days), in accordance with French recommendations [9]. Overall, we report better follow-up for infants born to mothers with confirmed toxoplasmosis, other than for treatment, which was poorly documented (76% missing data). Exploration of the immune responses of the mothers and infants by immunoblotting can improve the accuracy of CT diagnosis, thereby facilitating early management [32]. Such screening was not available in Mayotte at the time of our study, but was implemented soon after.

Given the seriousness of this disease, several countries have, like France, implemented preventive programmes. For example, in Italy, Denmark, and Germany, CT monitoring has been performed since 1997, 1999 and 2001, respectively [33]. However, in 2007, Denmark decided to stop its screening programme, because of the high costs relative to the low level of potential benefit. Other European countries, such as Lithuania, and Slovenia, consider CT to be a public health problem and deliver advice about its prevention, but have yet to establish an effective CT surveillance system. Austria and Brazil, like France, have toxoplasmosis screening programmes [34], but screening is not recommended in the United Kingdom and North America, where incidence is low [35–37]. The monthly screening of all seronegative women has been implemented in France since 1992, followed by the systematic declaration of CT cases. The prevention programme implemented in 2010 has proved effective in mainland France, decreasing the levels of epidemiological markers and rates of severe forms of toxoplasmosis [11]. Unfortunately, this programme is not fully applied in Mayotte, where serological screening is performed during pregnancy (when women attend prenatal consultations), but CT cases are not reported to the toxoplasmosis NRL (national reference laboratory). Several factors contribute to this situation: the frequent use of false identities by illegal immigrants (20% of pregnant women in Mayotte are thought to be illegal immigrants), the difficulties understanding French of most of the women and a lack of awareness of surveillance programmes among healthcare workers, together with challenging working conditions (insufficient numbers of healthcare staff) [38].

Our findings have important public health and social implications and should lead to the strengthening of educational programmes in Mayotte, for both healthcare practitioners and pregnant women. Specific cultural characteristics and practices should be considered as potentially important independent determinants in the planning of further counselling strategies, as a better understanding of complementary diagnostic examinations and the consequences of the disease would improve patient awareness and decrease opposition to specific management practices, including treatment and invasive procedures.

This study is the first to report epidemiological data on maternal and congenital toxoplasmosis management in Mayotte and highlights possibilities for improvements and many issues that remain to be addressed.

## Acknowledgments

We thank Dr. Guillaume Miltgen (UMR Processus Infectieux en Milieu Insulaire Tropical, CNRS 9192, INSERM U1187, IRD 249, Université de La Réunion, Saint-Denis, La Réunion, France) for comments on the manuscript and for his insights in review and editing.

## Author Contributions

**Conceptualization:** Laure Kamus, Christelle Vauloup-Fellous.

**Data curation:** Laure Kamus.

**Investigation:** Laure Kamus.

**Methodology:** Laure Kamus, Christelle Vauloup-Fellous.

**Resources:** Laure Kamus, Sophie Belec, Laurent Lambrecht, Soumeth Abasse, Sophie Olivier, Patrice Combe, Pierre-Emmanuel Bonnave.

**Visualization:** Christelle Vauloup-Fellous.

**Writing – original draft:** Laure Kamus, Christelle Vauloup-Fellous.

**Writing – review & editing:** Laure Kamus, Christelle Vauloup-Fellous.

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
