## [Decision Letter · Decision Letter 0]

14 Oct 2022

Dear Docteur KAMUS,

Thank you very much for submitting your manuscript "Maternal and congenital toxoplasmosis in Mayotte: prevalence, incidence and management" for consideration at PLOS Neglected Tropical Diseases. As with all papers reviewed by the journal, your manuscript was reviewed by members of the editorial board and by several independent reviewers. In light of the reviews (below this email), we would like to invite the resubmission of a significantly-revised version that takes into account the reviewers' comments. 

This is a well planned study and also goes well with the country's policy for updated information of prevalence of Toxoplasmosis in pregnant women in Mayotte (France territory). However, as highlighted by one of the reviewer, there is a lot of scope for language improvisation and particularly the in-house standardized abbreviations should be removed.

We cannot make any decision about publication until we have seen the revised manuscript and your response to the reviewers' comments. Your revised manuscript is also likely to be sent to reviewers for further evaluation.

Sincerely,

Sarman Singh, MD, FRSC, FRCP

Guest Editor

Charles Jaffe

Section Editor

This is a well planned study and also goes well with the country's policy for updated information of prevalence of Toxoplasmosis in pregnant women in Mayotte (France territory). However, as highlighted by one of the reviewer (the only good review), that there is a lot of scope for language improvisation and particularly the in-house standardized abbreviations should be removed.

Reviewer's Responses to Questions

**Key Review Criteria Required for Acceptance?**

**Methods**

-Are the objectives of the study clearly articulated with a clear testable hypothesis stated?

-Is the study design appropriate to address the stated objectives?

-Is the population clearly described and appropriate for the hypothesis being tested?

-Is the sample size sufficient to ensure adequate power to address the hypothesis being tested?

-Were correct statistical analysis used to support conclusions?

-Are there concerns about ethical or regulatory requirements being met?

Reviewer #1: satisfactory

Reviewer #2: -Are the objectives of the study clearly articulated with a clear testable hypothesis stated : yes 

-Is the study design appropriate to address the stated objectives : yes

-Is the population clearly described and appropriate for the hypothesis being tested : 

See comments below on the need to better define the population of pregnant women in terms of length of pregnancy monitored.

-Is the sample size sufficient to ensure adequate power to address the hypothesis being tested : yes

-Were correct statistical analysis used to support conclusions?

-Are there concerns about ethical or regulatory requirements being met? : no

Reviewer #3: -Are the objectives of the study clearly articulated with a clear testable hypothesis stated? Yes

-Is the study design appropriate to address the stated objectives? Yes

-Is the population clearly described and appropriate for the hypothesis being tested? yes

-Is the sample size sufficient to ensure adequate power to address the hypothesis being tested? Yes

-Were correct statistical analysis used to support conclusions? Yes

-Are there concerns about ethical or regulatory requirements being met? Authors can provide ethical clearance or permission (though this is a retrospective study)

**Results**

-Does the analysis presented match the analysis plan?

-Are the results clearly and completely presented?

-Are the figures (Tables, Images) of sufficient quality for clarity?

Reviewer #1: The manuscript is interesting; the methods are satisfactory and results are important. 

- Please denote the sex of infants born to mothers with toxo. It is important because maternal inflammation could have more adverse impacts on males. 

- Please add clinical information (symptoms and outcomes) of infected mothers or infants if available.

Reviewer #2: See below for comments on the results section.

Reviewer #3: -Does the analysis presented match the analysis plan? Yes

-Are the results clearly and completely presented? Yes

-Are the figures (Tables, Images) of sufficient quality for clarity? Yes. In addition, an infographic figure of the overall study is recommended.

**Conclusions**

-Are the conclusions supported by the data presented?

-Are the limitations of analysis clearly described?

-Do the authors discuss how these data can be helpful to advance our understanding of the topic under study?

-Is public health relevance addressed?

Reviewer #1: satisfactory

Reviewer #2: See below for comments on statement provided in the discussion

Reviewer #3: - Are the conclusions supported by the data presented? Yes

-Are the limitations of analysis clearly described? Yes

-Do the authors discuss how these data can be helpful to advance our understanding of the topic under study? Yes

-Is public health relevance addressed? Yes

**Editorial and Data Presentation Modifications?**

Reviewer #1: (No Response)

Reviewer #2: General comments: 

It is current practice to use “Toxoplasma infection” instead of “Toxoplasmosis” for cases that are not patent. 

“Tx” is not a standard abbreviation for “Toxoplasma infection” or “Toxoplasma gondii” and no single abbreviation should be used for the infection and the agent causing it. (When referring to IgG we would say “anti Toxoplasma” IgG and not “toxoplasmosis IgG”) Its extensive use throughout the manuscript largely decreases its readability. Congenital Toxoplasma infection can be abridged with CT, which will reduce the number of characters to such extend as to allow the use of "infection" instead of "PI" in pregnant women. In the context of the study, it is obvious that primary infections are the events of interest rather than reactivations

The abbreviation CPI, PCI, PPI, NPI should be simplified. 

Introduction 

The use of brackets should be avoided for important information about treatment and follow up. Hygiene recommendations are well placed since 1983. The French surveillance programme does not include registering maternal infections, but only CT. The exact dates of the study period do not need to be given here (there are given in the methods section). 

Material and methods

Line 113-115: It is standard practice when dealing with a population of pregnant women to define the inclusion criteria based on their dates of pregnancy or of delivery. 

It does not seem to have been the criteria used here. If the entry point for patients' inclusion was the existence of at least one serological test for anti Toxoplasma IgG and IgM collected within a given period, it should be stated more clearly. "All pregnant women for whom at least one anti serological test for anti toxoplasma IgG and IgM was performed between January 2017 and August 2019" were included regardless of the gestational age at sampling.

Lines 117-127: as mentioned earlier, replace TX with Toxoplasma

Lines 130-156: Tx can be deleted throughout because of the mention of the tests: ex: Vidas Toxo Avidity. The decision to consider equivocal IgG results as negative has to be defined here and not in the results section. 

Lines 135-137; information about doubling of IgG titres.. should be removed (repetition with the next paragraph)

Lines 141 156: this paragraph is very hard to read because of the many repeated words. It also seems that some situations are missing (criteria for excluding an infection during pregnancy). 

I would suggest limiting to three situations that involve different management.

No infection during pregnancy : past infection (Early and stable IgG) or still negative at delivery

Proven Infection during pregnancy : true seroconversion or doubling of IgG in the context of high titters

All other situations including : late positive IgG, insufficient screening until delivery. 

How prevalence was measured in pregnant women should be mentioned here = proportion of pregnant women with IgG above the cut off at their first test

The same goes for the incidence of Toxoplasma infection in pregnancy. It should be stated that only true seroconversions and proven profils of acute infection were taken into account to estimate incidence. What was the denominator used: what is the total number of pregnant women regardless of the length of pregnancy covered by the screening, or the total number of month of pregnancy covered by the screening, if no tests were performed at delivery.

How incidence of Congenital toxoplasmosis was computed should also be stated here rather on line 240 and 242

Results

The results section is overall difficult to read because of the information that should be given in the material and methods section, because confirmed cases of infection tend to be presented in the same length of details as non confirmed cases (including in table 1 and 2), and because of the use of many abbreviations. Figure 1 is especially hard to read because of the abbreviations. CPI, NEPI and PCI should best be written in clear, and because it mixes criteria for the different profiles, findings at the first test, and findings of follow up. I would suggested limiting figure 1 to findings at the first test. 

Tables are also difficult to read because of the repetition of the column denominator on each line. "%" does not need to be repeated on each line if define in the column head. "Primary" can be excluded from Table 1. 

The number of tests per pregnancy should be given at the beginning of the section, and how many seronegative patients were tested until delivery. It is a key information, to estimate the likelihood of detecting an acute infection during the whole course of pregnancy. 

Line 173: Prevalence of maternal toxoplasmosis should be replaced with Prevalence of IgG and IgM in pregnant women. It should be stated that it was measured on the first test, performed on average at x weeks of pregnancy (+ range)

Discussion

Line 265. The decrease in prevalence rather reflects epidemiological changes and rarer opportunities of encounters with the parasites as they are also detected in men. 

Line 288. This should be reworded because the risk for a single non immune pregnant woman of getting infected does not depend on the prevalence among pregnant women. It also should be clearly stated that incidence among pregnant women is only based on estimates, based on modelling. 

Line 340. Austria monitors the total number of children recognized as infected. 

Line 344. 1978 was the year when pre-marrital testing was implemented. Monthly re-testing was implemented in 1992. The French Register for Congenital toxoplasmosis was only started years later. No prevention program has been implemented in 2010. This paragraph has to be rewritten.

Reviewer #3: (No Response)

**Summary and General Comments**

Reviewer #1: Overall, the manuscript is interesting; the methods are satisfactory and results are important. However, it needs to add some information of the patients:

- Please denote the sex of infants born to mothers with toxoplasmosis. It is important because maternal inflammation could have more adverse impacts on males. 

- Please add clinical information (symptoms and outcomes) of infected mothers or infants if available.

Reviewer #2: This study provides valuable information 

- on the risks of maternal and congenital Toxoplasma infections in a territory and geographical area for which information is lacking

- and on the lack of compliance with the screening and management recommendation that apply to all French territories.

It clearly illustrates the overall importance of supporting the setting up of preventive actions by appropriate organization and communication, and of monitoring its long term benefits. 

Care should be taken to make the information easier to read and understand by reducing the number of abbreviation and trimming unnecessary words. Strengthening the material and methods section by providing all required information will help the reader and make the results more visible.

Reviewer #3: The study on "Maternal and congenital toxoplasmosis in Mayotte: prevalence, incidence and management" from France in generally interesting and provide insightful data and future direction.

Strengths:

For epidemiological stand point, this study provides important public health and social implications to strengthening of educational programs (counselling strategies, complementary diagnostic examinations, and awareness) for both healthcare practitioners and pregnant women in Mayotte-France.

Weakness:

one of the limitations of the study is the inconsistency of serological follow up in pregnant women. My other comments are on the manuscript file (highlighted in yellow)

PLOS authors have the option to publish the peer review history of their article (what does this mean?). If published, this will include your full peer review and any attached files.

Reviewer #1: No

Reviewer #2: No

Reviewer #3: No
---

## [Decision Letter · Decision Letter 1]

27 Feb 2023

Dear Docteur KAMUS,

We are pleased to inform you that your manuscript 'Maternal and congenital toxoplasmosis in Mayotte: prevalence, incidence and management' has been provisionally accepted for publication in PLOS Neglected Tropical Diseases.

Best regards,

Sarman Singh, MD, FRSC, FRCP

Academic Editor

Charles Jaffe

Section Editor

Reviewer's Responses to Questions

**Key Review Criteria Required for Acceptance?**

**Methods**

-Are the objectives of the study clearly articulated with a clear testable hypothesis stated?

-Is the study design appropriate to address the stated objectives?

-Is the population clearly described and appropriate for the hypothesis being tested?

-Is the sample size sufficient to ensure adequate power to address the hypothesis being tested?

-Were correct statistical analysis used to support conclusions?

-Are there concerns about ethical or regulatory requirements being met?

Reviewer #1: (No Response)

Reviewer #3: -

**Results**

-Does the analysis presented match the analysis plan?

-Are the results clearly and completely presented?

-Are the figures (Tables, Images) of sufficient quality for clarity?

Reviewer #1: (No Response)

Reviewer #3: -

**Conclusions**

-Are the conclusions supported by the data presented?

-Are the limitations of analysis clearly described?

-Do the authors discuss how these data can be helpful to advance our understanding of the topic under study?

-Is public health relevance addressed?

Reviewer #1: (No Response)

Reviewer #3: -

**Editorial and Data Presentation Modifications?**

Reviewer #1: (No Response)

Reviewer #3: -

**Summary and General Comments**

Reviewer #1: (No Response)

Reviewer #3: -

PLOS authors have the option to publish the peer review history of their article (what does this mean?). If published, this will include your full peer review and any attached files.

Reviewer #1: No

Reviewer #3: No

---

## [Editor Report · Acceptance letter]

16 Mar 2023

Dear Docteur Kamus,

We are delighted to inform you that your manuscript, "Maternal and congenital toxoplasmosis in Mayotte: prevalence, incidence and management," has been formally accepted for publication in PLOS Neglected Tropical Diseases.

Best regards,

Shaden Kamhawi

co-Editor-in-Chief

Paul Brindley

co-Editor-in-Chief
